# Potential of Halophytes-Associated Microbes for the Phytoremediation of Metal-Polluted Saline Soils

**Pauline Bonaventure ***, **Linda Guentas ***, **Valérie Burtet-Sarramegna *** and **Hamid Amir**

Institute of Exact and Applied Sciences, University of New Caledonia, PCP3+R8M, Ave James Cook, Noumea 98800, New Caledonia

\* Correspondence: pauline.bonaventure@outlook.fr (P.B.); linda.guentas@unc.nc (L.G.); valerie.sarramegna@unc.nc (V.B.-S.); Tel.: +687-971940 (P.B.)

**Abstract:** Saline ecosystems are often the target of spills and releases of pollutants such as metals, as many industrial companies settle in or around these areas. Metal pollution is a major threat for humans and ecosystems. In line with sustainable development, nature-based solutions and biological tools such as phytoremediation offer eco-friendly and low-cost solutions to remove metals or limit their spread in the environment. Many plant-growth-promoting (PGP) effects are frequently prospected in plant-associated microbes such as the production of auxins, siderophores, or extracellular polymeric substances to enhance phytoremediation. Halophytes are nowadays presented as good phytoremediators for metal-contaminated saline environments such as coastal regions, but little is known about the potential of their associated microbes in the bioaugmentation of this technique. Here, we review the studies that focused on halophytes-associated microbes and their plant-growth-promotion capacities. Moreover, we discuss the limitation and applicability of bioaugmented phytoremediation in saline ecosystems.

**Keywords:** halophytes; PGP microbes; metal phytoremediation; bacteria; fungi

## 1. Introduction

Metal pollution related to urbanization and industrialization and associated with a rapid economic growth has increased in the coastal areas during the last decades [1–3]. Anthropogenic activities such as mining, textile industries, and agriculture—and, more generally, soil degradation—lead to the transfer of metals-rich particles to the coastal environments. About 80% of the pollutants from human activities are introduced into coastal environments, contaminating waters, sediments, and biological systems [1,4]. Indeed, salt marshes and other saline ecosystems are often the target for the dumping of toxic pollutants and the establishment of landfills. The reason for this is that for, a long time, they were considered to be areas of little interest, especially due to the absence of glycophytes, on which agriculture is mainly based [5]. According to Andersen et al. [6], 75–96% of European seas are contaminated with metals.

Many coastal regions are highly polluted. There are almost 620,000 km of coastlines worldwide, and over one-third of the total human population lives within 100 km of the coast. The intertidal ecosystems are very important for terrestrial and marine organisms, whose life cycles are often partially or entirely dependent them. Although coastal environments act as buffering zones at the land–sea interface, limiting the spread of contaminants to the ocean, contamination levels are such that the retention effects of these ecosystems are often insufficient, and metals still accumulate in the food chain of surrounding ecosystems [7]. Affecting marine organisms [3,8,9], invertebrates [3,10], plants [11], microorganisms [12], and representing a threat for human health [3,13,14], metal pollution in coastal areas has become a matter of crucial major concern.

Over the last decades, the removal of metals in coastal wetlands with biological remediation means has thus received much attention from scientists, especially for mangroves,

estuaries, salt marshes, and forests [15,16]. Indeed, biological solutions have received interest since the costs and eco-friendly solutions needed to remediate metal contaminations have increased, especially in highly industrialized areas of poor and in-development countries. According to Origo et al. [17], interest for phytoremediation strategies has only thrived during the past 20 years because of growing commercial issues, especially in the Northern American countries. Phytoextraction of metals is a phytoremediation technique that consists of absorbing metals from the soil, using plants that preferentially accumulate these elements in their aerial parts [18–20]. The metals can potentially be recovered from plant biomass in forms that can be subsequently further exploited for several industrial purposes [21]. On the other hand, metal phytostabilization uses plants that reduce "the mobility and bioavailability of metals within the surroundings, thereby preventing their migration to groundwater or their entry into the food chain" [20]. Origo et al. [17] put forth the engagement for sustainable development when developing such biotechnologies. Westphal and Isebrands [22] highlighted the psychological and social implications of phytoremediation in the brownfield redevelopment. Phytoremediation and, more especially, green spaces development have shown positive effects on people in terms of medical needs, work productivity, and social relationships. Moreover, biological remediation techniques are cost-effective solutions and are increasingly recommended in scientific publications. Indeed, the costs of phytoremediation applications are particularly low compared to conventional techniques [23]. For Chen and Li [24], the costs/effectiveness ratio of phytoremediation was higher than in the case of excavation and disposal or soil washing only for low contamination levels for a duration of 5 years of treatment. These authors recommend combining phytoextraction and physicochemical techniques to accelerate and improve the process. However, microbes could also accelerate and improve the phytoremediation yield [25,26]. Many plant-associated microbes are known to increase plant growth and stress resistance. Microbial remediation, the process for removal of environmental contaminants with the use of microorganisms [27,28], can be combined with phytoremediation. Indeed, microbe-assisted phytoremediation techniques involving bacteria and/or fungi and plants have been proposed as a cost-effective and reliable approach to improve the intrinsic bioaccumulation capacities in plants [7,29,30]. Bioleaching, biosorption, bioaccumulation, bioprecipitation, and biotransformation represent the main strategies evolved by microbes in order to cope with contaminants from the soils. Hence, microbes are often prospected as interesting potential candidates in bioremediation [21].

The abilities of halophytes to cope with high levels of metals have been widely studied [31–37]. Indeed, by using the search item "halophyte OR halophytes AND (metal OR metals)" on Scholar Google (http://scholar.google.com, accessed on 12 March 2023), we found 27,400 results. Representing 1–2% of the world's flora [38,39], this plant group harbors numerous interesting properties for several application fields (including food, pharmacology, and energy) [40], and some species are already recommended for the phytostabilization or phytoextraction of organic and inorganic pollutants. In the 2000s, researchers began to study halophytes-associated microbes for their bioremediation potential, and the number of publications on their use as plant-inoculants only increased in the late 2000s/early 2010s, but studies mainly focused on saline-affected soils rather than metal-contaminated saline soils. However, numerous papers dedicated to the restoration of agricultural soil functions affected by salt show interesting plant-growth-promoting (PGP) properties of halotolerant microbes that could also benefit plants under metal stress [41–44]. Owing to their halophilic/halotolerant adaptations, these microbes unfold multiple PGP effects and metal-tolerance strategies [45–47], thus emphasizing their potential in metal-contaminated saline soils' treatments [15,48–51]. Nevertheless, microbe-assisted phytoremediation applied to metal-contaminated coastal environments still lack in knowledge to be properly used, despite the warnings about the impacts of the contaminants spread in the oceans. Furthermore, despite a substantial number of experiments conducted for this purpose, no specific review has, to our knowledge, focused on the use of halophyte-associated microbes as inoculants to improve the effectiveness of halophyte phytoremediation of metal

contaminated saline soils. Indeed, research on PGP microbes was mainly reviewed in the context of metal contamination in non-saline sites [27,52–58]. Some reviews concerned applications to salt-affected sites by using halotolerant bacteria for agriculture [41–44,59,60] but did not develop their potential for the improvement of metal phytoremediation.

In this review, we focus on the current research on (i) the potential of halophytes and their associated microbes in the phytoremediation of metals; (ii) the mechanisms employed by these halotolerant microbes to deal with metals and to improve plant growth; (iii) how using a microbial inoculum combining bacteria and fungi can benefit plants and (iv) the in situ applications of microbe-assisted phytoremediation, including their limitations and benefits. As several publications described the PGP effects commonly screened in microbes, the most common ones will be briefly presented here to better contextualize their implication in the optimization of metal-phytoremediation techniques applied to saline environments. It is obviously the first review that concentrates on bioremediation applied to metal-contaminated saline environments that simultaneously used the remediation potential of halophilic plants and microbes. Given the magnitude of metal pollution on the world's coasts, this work provides reasoned perspectives on the application of microorganisms in phytomanagement techniques and highlights the obscure points of this type of applications that remain to be elucidated in scientific research.

## 2. Potential of Halophytes and Halophytes-Associated Microorganisms for the Bioremediation of Metals in Contaminated Saline Environments

### 2.1. Potential of Halophytes

Salt stress induces plant responses similar to those caused by metals. Indeed, as the main stressful driver for species living in saline areas [61], salt stress has led to strong adaptations based on the exclusion, compartmentation, and excretion of ions and oxidative defense system responses that counteract the effects of reactive oxygen species (ROS) over-produced during stresses such as salt or metal stress [62]. Morphological adaptations to salinity can help plants to tolerate metals within tissues. Succulence, currently considered to be a halophytic or xerophytic trait, could also alleviate growth injuries due to metal stress without reducing metal bioaccumulation [63,64]. Succulence can indeed improve the water-use efficiency, thus enhancing photosynthesis under salt stress [65]. Halophytes deal with salt stress owing to three main strategies: salt excluders, where plants limit toxic ions absorption by roots; salt includers or accumulators, which are able to tolerate high NaCl concentrations, especially in their aerial parts; and salt excretors, which possess salt glands that excrete the absorbed salt at the leaf surface [39,66]. Metal phytoremediation requires the use of plants' resistant to contaminants able to reduce the bioavailability of metals or able to tolerate metals in shoots, the harvestable parts of plants [54]. Plants that preferentially translocate metals in shoots with a high specificity for the metal(s) that are easy to grow and offer the possibility of several harvesting seasons and are perfect candidates for phytoextraction, but they are not so easy to find. According to Sarwar et al. [54], phytoextraction is more suitable than phytostabilization, as it allows us to definitely remove contaminants from the soil. However, phytoextraction applications must combine several characteristics, particularly a high metal translocation rate from roots to shoots, and ideally a hyperaccumulating ability, as recommended by the authors. Most of halophytes accumulate metals in the roots, and only rare halophyte species are considered to be metal hyperaccumulators [67,68]. Anyhow, the choice of halophyte species that rather belong to salt includers, ideally capable of accumulating metals in shoots and in roots, is relevant in the development of phytoremediation applications [54,69]. Despite variations in their mechanisms of adaptation to salinity, all halophytes can use exclusion and inclusion mechanisms [70]. Depending on their salt and metal toxicity thresholds, these plants limit or tolerate metals in their internal parts. However, some other parameters can influence metal toxicity and thus limit the tolerance mechanisms. Metals' bioavailability and uptake by the plants greatly depend on salinity [1,64]. In the study of Wali et al. [71], adding 200 mM NaCl to *Sesuvium portulacastrum* plants alleviated the Cd stress by reducing the Cd concentrations

in the shoots. Moreover, the global accumulated amounts of Cd were unchanged, and the growth was enhanced compared to the plants without NaCl. According to Bai et al. [2], the distribution patterns of metals in soil is influenced by plant communities because plant roots can influence the stabilization or release of metals. Some halophytes are described as relevant candidates of phytoremediation, such as *Sesuvium portulacastrum* [72,73], *Atriplex halimus* [74], *Atriplex hortensis* [75,76], or *Climacoptera crassa* [77]. Specific species can also accumulate large amounts of metals. For instance, *Arthrocnemum* can accumulate 724 mg/kg DW of Pb [78]. *Salsola kali*, a desert halophyte, can bear 2075 mg/kg Cd in the stems and 2016 mg/kg Cd in the leaves [79] and may be classified as "obligate hyperaccumulators" according to Aziz and Mujeeb [68]. The desert halophylic *Suaeda fructicosa* can accumulate 1379 mg/kg Cr and a maximum of 13,246 mg/kg Na in the shoots, while soil concentrations were 19.5 mg/kg [80]. In practice, the development of a phytomanagement strategy should require the definition of where metals will concentrate according to the field specificities and how to manage the remediated metals during or after the phytoremediation process. From this point of view, most of the publications evaluating the potential of halophyte species in metal phytoremediation distinguish the location(s) where metals are mostly accumulated in the plant, as this generally determines the indication for phytoextraction or phytostabilization [18]. Manousaki et al. [34] mentioned another faculty of some salt excretors to proceed with phytoexcretion in supplement of phytoextraction, e.g., as with *Tamarix smyrnensis*, but this faculty should be further investigated. Naikoo et al. [36] presented a classification of the different halophyte species owing to the removal strategy that are frequently recommended for the phytoremediation of metals, especially in India. For instance, species such as *Atriplex halimus* can be indicated for phytoextraction of Cd, Pb, Mg, and Zn, while *Arthrocnemum macrostachyum* can be applied to Cd phytostabilization.

### 2.2. Potential of Halotolerant and Halophilic Microbes in the Mediation of Metal Stress in Plants

Microbes in the rhizosphere can drastically influence soil parameters in soils [21]. According to Mishra et al. [27], microbial activities can limit the bioavailability of metals for the ecosystem and thus control the absorption of metals by accumulating plants. Moreover, microbial interactions with plants include a communication system made of an exchange of several secondary metabolites that will thus lead to changes in plant responses to stress [53,81]. Hence, the microbiome must also be considered when engineering phytoremediation techniques.

According to Qin et al. [59], "any community of root-associated microorganisms would be dominated by plant-growth-promoting rhizobacteria (PGPR) and endophytic bacteria". The number of publications describing the mechanisms related to PGP microbes in double stressful conditions, namely with high metal and salt concentrations, is relatively low. Indeed, when using the items "halophyte" AND "PGP" AND "metals" OR "metal" on Scholar Google (https://scholar.google.com/, accessed on 12 March 2023), we found 684 results.

In plants, salinity exerts a selective pressure that could induce resistance to other stresses, such as metal toxicity [82], and the same process could have happened for microbial communities. Moreover, Mohapatra et al. [83] suggested that most of the bacteria isolated from saline environments have to cope with multiple fluctuating abiotic parameters, such as temperature, pH, and salinity, and are consequently likely to tolerate other stresses, an idea which is also supported by Yuan et al. [84]. Dealing with salinity implies adaptations related to proteins' stability [85], lipid composition and membrane fluidity [15,86], ion homeostasis, and the transport system to regulate Na concentrations in cells, the production of compatible solutes [15]. Moreover, microbes have to face a fluctuant salinity in tidal environments throughout the day and the year. Vauclare et al. [87] emphasized that the good adaptability of some non-extreme halophilic and halotolerant microbes to several stressors could be linked to a repeated change of the salinity level, namely occurring in coastal areas, leading to an adaptive process on their proteins that make them stable under diverse salinities. Voica et al. [51] asserted that candidates for metal bioremediation can

be found in both halophilic and halotolerant bacteria and archaea. Several publications showed a potential in metal bioremediation with halotolerant [45,46,88,89] and halophilic microbes [89–93].

According to Yuan et al. [84], the halophyte microbiome could be enriched in genes involved in plant salt resistance and PGP capacities. These authors showed that the microbiome of *Suaeda salsa* was enriched in bacteria with genes related to salt stress acclimatization (ABC transporters), nutrient acquisition (phosphatase, pyrroloquinoline-quinone synthase, nitrogen fixation protein, nitronate monooxygenase, formamidase, and nitrite reductase), and competitive root colonization (site-specific recombinase/integrase and NADH dehydrogenase). Positive effects of halophytes-associated microbes on plant growth and salt resistance have also been shown, especially for some plant genera: *Salicornia* [30,94–96], *Limonium* [97], *Arthrocnemum* [45], *Spartina* [46,98], and *Sulla* [99]. In view of the particularly fluctuant environmental conditions of the seashore areas, the salt adaptations of halophytes-associated microbes are linked to multiple PGP properties that are relevant in the bioaugmentation of metal phytoremediation applied to such sites (Figure 1).

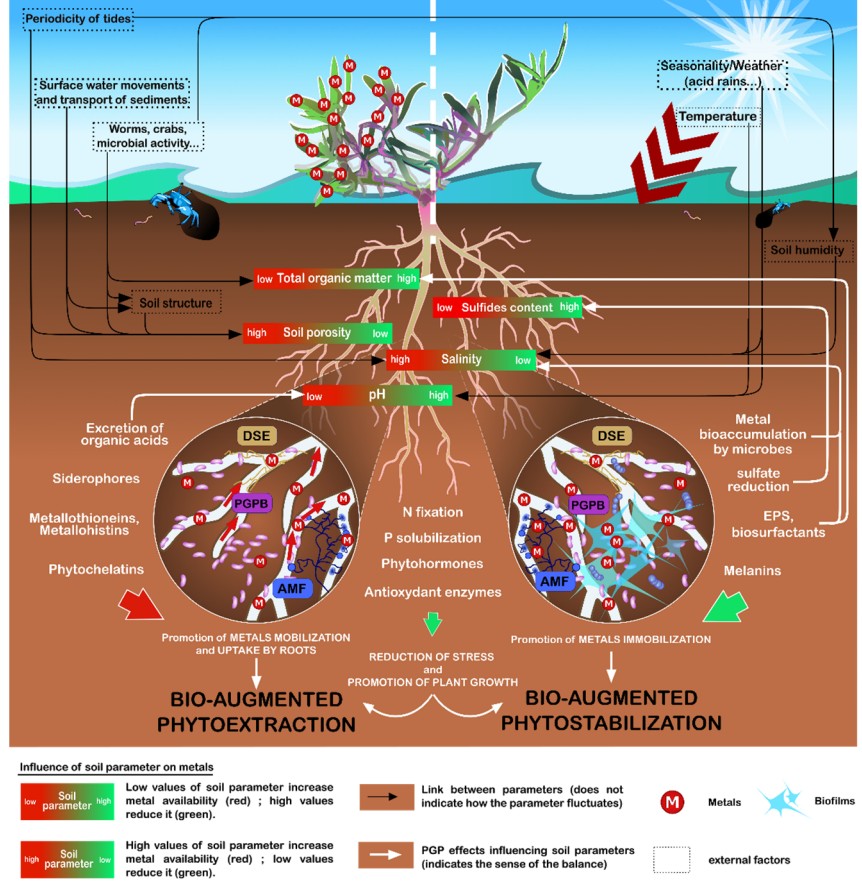

**Figure 1.** Factors influencing the phytoextraction and the phytostabilization by halophytes and their associated microbes in estuarine metal-contaminated saline soils. The bioavailability of metals determines the phytoremediation yield and depends on parameters such as pH, salinity, the total organic matter, and soil porosity. External factors framed in dotted line are specific of coastal environments and influence these parameters. On the left side, microbes favor the metal phytoextraction and translocation by enhancing metals mobilization via the production organic acids and metal-affine molecules. On the right side, microbes help plants to stabilize metals via their bioaccumulation, the production of metals immobilizing agents (extracellular polymeric substances and melanins), and the precipitation of metals by sulfate-reducing bacteria. In the middle, microbes can enhance plant growth by producing phytohormones, enhancing nutrients' availability, and reducing oxidative stress.

### 3. Selection of Microbes for the Phytoremediation of Metal Contaminated Saline Soils

*3.1. Frequently Investigated PGP Properties*

3.1.1. Production of Indole-3-Acetic Acid

Diverse PGP effects that modulate plant growth and help plants to tolerate salt and/or metal stress are frequently investigated, namely the production of several phytohormones: indole-3-acetic acid (IAA), abscisic acid (ABA), cytokinins (Cyt), gibberellic acids (GAs), or jasmonic acid (JAs) [42,56,59,92,100]. IAA is the most frequently tested.

Elevated metal concentrations induce ethylene production in plants, leading to root growth restriction, such as the decrease of root hair number, and roots branching and elongation [39,101]. Ethylene acts synergistically with IAA and the underlying mechanisms are nowadays well described [102]. IAA is often produced by bacteria and fungi [103–105]. Though its role for microorganisms remains unclear, IAA is considered as one of the most common phytohormones involved in biochemical and biological pathways of plant development such as cell enlargement and division, tissue differentiation, and responses to light and gravity [106]. Many studies have demonstrated the interest of treating plants with IAA-producing bacterial strains and identified the implication of this phytohormone in the improvement of several growth parameters and metal-bioaccumulation capacities [100,107]. Moreover, IAA-producing microbes can also enhance nutrition; increase the number of root hairs, rooting and root and shoot elongation, leaf area and number, and germination rate; and can also limit pathogen aggressions [107–110]. Bianco and Defez [111] showed that, under 0.3 M NaCl in the growth solution, inoculating *Medicago trunculata* with an IAA-overproducing *Sinorhizobium meliloti* strain can lead to higher internal proline contents, produced as an osmolyte, and enhance the activity of antioxidant enzymes compared with *M. trunculata* plants inoculated with the wild-type strain. However, the inoculation on a halophyte species was not evaluated. The glycophyte *Arachis hypogaea* inoculated with a *Bacillus licheniformis* strain isolated from the halophyte *Suaeda fruticosa* improved biomass, total length, and root length in the presence of NaCl [112]. The in vitro PGP characterization revealed the ability of this strain to produce IAA, among other PGP features. Inoculating plants with IAA producers can improve growth but does not necessarily lead to an increased metal accumulation by plants. For instance, Ikram et al. [113] showed that an IAA-producing *Penicillium roqueforti* strain isolated from the halophyte *Solanum surattense* could improve wheat growth and limited the absorption of Ni, Cd, Cu, Zn, and Pb in a metal-polluted soil compared with the non-inoculated plant.

3.1.2. Production of 1-Aminocyclopropane-1carboxylate Deaminase

The production and activity of the 1-aminocyclopropane-1-carboxylate deaminase (ACCD) is also often prospected [43,114–119]. The ACCD is intrinsically linked to the presence of IAA [107]. Indeed, the excessive production of IAA can limit root growth because of the production of 1-aminocyclopropane-1-carboxylate (ACC) and its transformation into ethylene [120]. The production of ACCD by microbes enables the plant to regulate the amount of ACC in the plant cells by hydrolyzing it into $\alpha$-ketobutyrate and ammonia, which may be furthermore a potential source of nitrogen for plants and microbes. This enzyme thus has a double positive effect for the plant: (1) reducing ethylene content responsible for the inhibition of root growth and (2) enhancing the plant nutrition [107,119]. In the study of Siddikee et al. [121], 25 of the 36 tested strains isolated from the rhizosphere of some halophytes had the ability to produce ACCD, and this activity varied among strains. The strains unfold diverse PGP features, and the inoculation of Canola under salt stress revealed that most of the strains could significantly enhance the root length and the total plant dry biomass.

3.1.3. Nutrition Improvement

Phosphate-solubilizing halophilic/halotolerant microbes are not rare. In coastal environments, phosphorus can be in the mineral ($HPO_4^{2-}$) or organic form (monophosphate esters, nucleotides, and their derivatives, vitamins, phosphonates, and humic acids). The

balance between these forms of phosphorus indirectly depends on oxygen distribution changes that periodically appear in tidal environments [122]. Moreover, saline environments such as tidal flats have a higher binding capacity for inorganic P because of their richness in Fe oxides [122]. Thus, solubilizing inorganic phosphate capacities of rhizospheric microbes can help salt marsh plants to extract available phosphorus. Goswami et al. [112] screened the PGP characteristics of *Bacillus licheniformis* strain previously mentioned, and also found that this strain could solubilize phosphate. It was also the case for 23 of the 85 rhizobacterial strains isolated from the rhizosphere of *S. fructicosa*, and 7 combined this PGP effect with the production of IAA. When exposed to 15% NaCl, 1 mM Cd, 0.7 mM Ni, 0.04 mM Hg, and 0.03 mM Ag, three strains of *Halobacillus* and *Halomonas* isolated from plant mangrove species were also capable of phosphate solubilization [47]. These strains could improve the root growth of *Sesuvium portulacastrum* watered with a 2% NaCl solution and cultivated in a metal-polluted sandy soil originating from coastal habitats.

Atmospheric nitrogen fixation is another PGP effect commonly screened in bioremediation techniques. In the study of Ozawa et al. [123], beneficial effects of inoculation on growth parameters of *Salicornia europea* were obtained under 0.2 to 0.3 M NaCl with a nitrogen-fixing rhizobacteria belonging to *Pseudomonas* genus. The inoculated plants had higher concentrations of total N, chlorophylls, and $Na^+$ and $K^+$ in the shoots. However, no significant differences were observed on fresh and dry weight of the shoots.

The faculty of producing siderophores helps plants in the iron nutrition and metal uptake. Siderophores exist under diverse molecular forms. According to Johnstone and Nolan [124], some siderophores easily bind other metals, such as Cu, Zn, Mn, or Cr, according to the molecule composition and structure. For instance, Mallick et al. [92] isolated two halophilic bacteria of *Kocuria flava* and *Bacillus vietnamensis* that alleviate As stress in plants owing to their PGP properties. The two strains were able to produce siderophores.

### 3.1.4. Multiple PGP Effects in Interaction with Metal Stress

Navarro-Torre et al. [45] screened PGP properties (nitrogen fixation, siderophores production, IAA production, and biofilm formation), and salt and metal resistance of bacterial strains isolated from the roots of the halophyte *Arthrocnemum macrostachyum* and belonging to *Halomonas*, *Kushneria*, *Micrococcus*, *Bacillus*, *Vibrio*, *Pseudoalteromonas*, and *Staphylococcus* genera. All of the strains showed at least one PGP property, but the presence of Cd had, in general, a negative impact, especially on the ability to fix atmospheric nitrogen. On the contrary, the production of IAA and the solubilization of phosphates were enhanced in presence of As and Cd, respectively. From this study, endophytic bacteria were the most resistant to metals. Another survey on *Spartina maritima* plants reported interesting PGP characteristics of rhizospheric-halotolerant bacteria isolated from the Tinto River estuary in Spain that is contaminated by Cu, As, Cd, Cu, Ni, Pb, and Zn: biofilm formation, siderophores production, IAA production, ACCD activity, phosphate solubilization, and nitrogen fixation [125]. PGP features were screened with and without Cu. The presence of high levels of metals decreased the number of strains exhibiting PGP features, except for the ability of forming biofilms. They selected the four most performing ones and inoculated *Medicago sativa* seeds to evaluate their effects on seed germination and root elongation. The results showed that all bacterial strains could significantly enhance the root biomass in absence of Cu, but only three of the four bacterial strains have the same effect in the presence of Cu. In another study [126], the same authors used a consortium of these four bacterial strains to inoculate *S. maritima* plants in contaminated soil from the Tinto River estuary. Inoculation had a significantly positive effect on the root growth and metal uptake, which increased by 19% for As, 65% for Cu, 40% for Pb, and 29% for Zn. The impact of PGP properties that are involved in augmented plant growth could also explain a better absorption and accumulation of metals because of greater storage of ions in relation to a greater plant biomass [30]. In the region of the Tinto and Odiel Rivers in Southern Spain, one of the most polluted areas of the world, bacteria from *Spartina densiflora*'s rhizosphere were tested for their contribution to growth in early steps

of plant development [127]. The germination rate was successfully improved when seeds were associated with a bacterial consortium composed of *Aeromonas aquariorum* SDT13, *Pseudomonas composti* SDT3, and *Bacillus* sp. SDT14. These bacteria showed several PGP features, namely phosphate solubilization, siderophores and IAA production, and nitrogen fixation. The consortium could increase seed germination and limited pathogenic fungal infection [127].

### 3.2. Metal Tolerance and Metal Inactivation Capacities

### 3.2.1. Metal Tolerance

To protect themselves against metals or salt stress, microorganisms deploy various mechanisms that could be beneficial to plants. In bacteria, the most commonly encountered metal-resistance mechanisms are metal sequestration by biopolymers [57,91,128], metal efflux via specific transporters, enzymatic detoxification [51,129], metal ion reduction, and intracellular storage [91,130].

Some bacteria can actively extract and accumulate metals. Amoozegar et al. [93] have evaluated the resistance to chromate, arsenate, tellurite, selenite, selenate, and biselenite of *Bacillus* strains isolated from Iranian saline soils. They showed that increasing NaCl concentrations from 5 to 15% incremented the metalloids' resistance of most of the strains. Similar results were obtained on moderate halophilic bacteria from the genus *Salinicoccus* isolated from saline soils in Iran: it was able to reduce tellurite in tellurium, extracting 75% of the total tellurite content in the medium containing 0.5 mM potassium tellurite and 10% NaCl [131]. They evaluated the influence of pH, temperature, and KCl and $Na_2SO_4$ concentrations on metal-extraction efficiency. The best results were reached with pH 7.5 at 35 °C and 1.5 M NaCl, 1 M KCl, and 0.5 M $Na_2SO_4$ were the optimal saline concentrations to remove potassium tellurite. Sowmya et al. [89] screened halophilic members of *Alcaligenes*, *Vibrio*, *Kurthia*, and *Staphylococcus* for their Cd- and Pb-resistance, via the evaluation of their minimum inhibitory concentrations (MICs) under saline conditions (5, 10, and 15% NaCl). The in vitro experiment showed that most selected isolates optimally removed Cd and Pb from the solution in presence of 10% NaCl. Mallick et al. [92] showed that *Kocuria flava* and *Bacillus vietnamensis* strains from the mangrove rhizosphere tolerate high concentrations of As (35 mM of arsenite for *Kocuria flava* and 20 mM for *Bacillus vietnamensis*).

As previously mentioned, salinity and metal stress lead to the production of reactive oxygen species (ROS), whose concentrations can be regulated by the activities of certain antioxidant enzymes. In this view, antioxidant activities are also commonly explored when screening microbes for their PGP features. Enzymes such as the superoxide dismutase (SOD), which transforms the radical $\bullet O_2^-$ into $H_2O_2$; the catalase (CAT), which transforms $H_2O_2$ into $H_2O$ and $O_2$; or the glutathione oxidase (GSH), which converts the gluthathione to glutathione disulfide, are involved in the detoxification of ROS and protect microbial cells against oxidative damages. Hence, measuring the activity of these enzymes helps in evaluating the stress levels and the level of tolerance to metal toxicity. Salt stress can additionally increase the activity of some antioxidant enzymes in halophilic bacteria, also inducing a higher metal tolerance, relating to similar mechanisms [62]. Among several PGP traits of a halophilic *Enterobacter* strain, it has been demonstrated that extreme salinity and alkaline conditions also increased the activities of the SOD, the CAT, the ACCD and the glutathione oxidase (GSH) [117].

### 3.2.2. Production of Exopolysaccharides

In the study of Mallick et al. previously mentioned [92], *Kocuria flava* and *Bacillus vietnamensis* strains were also able to adsorb As at their surface and accumulate this element intracellularly (2 mM As) under hypersaline conditions (from 0.5 to 2 M NaCl). The two strains were able to improve plant growth while reducing As accumulation in plants. For the authors, this could be due to their capacities of producing exopolysaccharides (EPSs). These molecules play an important role in the resistance of bacteria to stresses such as extreme pH, temperature, high salinity, and high metal contents [43]. EPSs surround cells by

forming a capsule or a viscous gangue in solution and can protect bacteria by neutralizing toxic $Na^+$ and metallic cations thanks to their negatively charged residues [132,133]. In the work by Ibrahim et al. [90], the production of EPS by a halophilic strain of *Halobacillus* isolated from Lake Qarun in Egypt was measured in the presence of 0 to 5 mM Cu, Zn, Cd, Pb, and Ni. Bacterial growth and EPS concentrations decreased with increasing metal concentrations, suggesting that the production of EPS could be related to the survival of the bacteria. The structure and biochemical properties of EPS and their contribution to the stabilization of metals have been studied and reviewed [7,44,57,90,115,128,134,135]. As it can reduce metal bioavailability and toxicity, the production of EPS is often considered to be a PGP effect and is frequently investigated in the development of bioaugmented phytoremediation techniques. These polymers are useful to control metal ions' flow into the roots. Their binding properties can vary along diverse parameters. Bhaskar and Bhosle [128] showed that the binding capacity for Pb and Cu of the EPSs produced by a *Marinobacter* strain was impacted by pH and salinity and differed between the two metal species. Metals and salt stress can stimulate their production. For instance, *Halomonas* bacteria isolated from the rhizosphere of *Avicennia marina* (a mangrove species) have been used to inoculate rice seedlings and showed PGP effects under salt and arsenic stress. Bacteria produced a greater amount of EPS with increasing salt or As concentrations [136]. Owing to their good metal binding properties, these biopolymers can be used in diverse forms to remediate metal contaminated soils and waters (application of activated sludge containing pure or mixed bacterial culture(s), application of dead biomass EPS, application of EPS immobilized in alginate or agar beads, or, more scarcely, application of chemically modified and enhanced EPS). The effectiveness of such treatments has been reviewed [133]. EPS are often found in bacterial biofilms as it influences the architecture and provides a stability of the biofilm [132]. Biofilm contains diverse molecules that bind metals, and this binding ability is another commonly investigated feature for the selection of metal-chelating compounds producing bacteria [45,46,92,125,137]. Paredes-Páliz et al. [46] tested different bacteria from the rhizosphere of *Spartina maritima* living in the metal-contaminated Odiel estuary. In the presence of As, Cu, Zn, and Pb, the Gram-negative strains of *Pantoea agglomerans* produced significantly higher amounts of biofilm than the Gram-positive *Bacillus aryabhattai*.

### 3.2.3. Arbuscular Mycorrhizal Fungi

Fungi also influence enzymatic systems under various stress. It is particularly the case of arbuscular mycorrhizal fungi (AMF). As reported by Juniper and Abbott [138], AMF are relatively sensitive to salinity, but with variations depending on strains. Alsadat Hashemi Nejad et al. [139] showed that plants inoculated with *Glomus mosseae* under Cd and salt stress had a lower proline content and SOD and CAT activities than in the control. Moreover, the inoculation increased shoot height and roots biomass, and according to the authors, this was linked to the ability of the fungi to reduce oxidative stress in the plant. Through other mechanisms, mycorrhizal colonization can facilitate metals' immobilization in the rhizosphere or inside roots. The AMF *Glomus geosporum* isolated from the halophyte *Aster tripolium*, whether sampled on polluted or non-polluted salt marshes, increased the accumulation of Cd and Cu in the roots, compared to the non-inoculated plants, but without enhancing plant growth [140]. Moreover, metal concentration can positively influence the AMF colonization. For instance, Suntornvongsagul et al. [141] showed that AMF colonization of the halophyte *Spartina patens* was higher in Ni-amended treatment than in the control.

### 3.2.4. Dark Septate Endophytes

Though the research interest on dark septate endophytes (DSEs) is still recent, they are nowadays reputed to be resistant to multiple stressors, readily colonizing disturbed ecosystems [142,143]. DSEs are a very diverse group of ascomycetes characterized by the ability of producing melanin, giving the appearance of brown hyphae [143,144]. Several

authors hypothesized that melanin could have a protective role against metal toxicity. Zhan et al. [145] demonstrated that the inhibition of the 1,8 dihydroxynaphthalene melanin synthesis by tricyclazole introduced in non-toxic concentrations and under Cd stress led to a reduction of growth and sporulation of the fungi *Exophiala pisciphila*, a Cd-tolerant DSE. However, the inhibition of melanin synthesis did not reduce the Cd bioaccumulation. Moreover, according to Priyadarshini et al. [146], melanins are molecules that can bind metals because of their chemical nature, i.e., rich in phenols, peptides, carbohydrates, fatty acids, and aliphatic hydrocarbons. It often forms deposits within the cell wall, thanks to chitin [147]. As reviewed by Singh et al. [148], fungal melanins can bind metallic cations. Oh et al. [149] extracted melanin from *Amorphotheca resinae* and showed good sorption properties toward Cu(II), Pb(II), Cd(II), and Zn(II). The extracted melanins from this latter species and *Aureobasidium pullulans* were also investigated for their binding properties with $Cu^{2+}$ [150]. The authors concluded that melanin could reduce toxicity and have a high affinity for $Cu^{2+}$. They suggested that this pigment could be used as a metal biosorbent. Furthermore, DSEs regulate some metal-resistance mechanisms of plants, such as the glutathione system [151]. DSEs can also deploy several PGP capacities. For instance, the study of PGP features of several strains of *Cadophora*, *Leptodontidium*, *Phialophora*, and *Phialocephala* revealed their capacity to produce IAA and volatile organic compounds that were supposed to help plant growth via gaseous exchanges [104]. Warda and Fortas [152] evaluated the phytoremediation capacities of five halophyte species, namely *Atriplex halimus*, *A. canescens*, *Suaeda fruticosa*, *Marrubium vulgare* and *Dittrichia viscosa*, from wetlands polluted by Cd, Cu, Zn, Pb, Ni, and Cr. They showed that fungal colonization by AMF and DSE helped to stabilize metals by sequestration in root apoplasm or by spore metal retention. AMF colonization was negatively correlated to the translocation of Cd, while DSE colonization was negatively correlated to the bioaccumulation of Cu and positively correlated to the translocation of Zn. Moreover, in 80% of plant species, an association between AMF and DSE was observed. Another study [153] showed that Louisiana marshes were colonized by AMF and diverse DSE, but DSE colonization was negatively correlated to AMF colonization.

### 3.2.5. Other Fungi

According to Liu et al. [154], a halophilic strain of *Aspergillus glaucus* possesses a protein (from the large ribosomal subunit 60 S) associated with salt resistance and conferring an improved tolerance to other stresses, such as drought and metal stress. In obligate halophilic fungal strains of different *Aspergillus* species and in *Sterigmatomyces halophilus*, the bioaccumulation of Cd, Cu, Fe, Mn, Pb, and Zn in their mycelium was moderate to high [48]. This study suggested that metal accumulation and resistance of these strains could be linked to their halophilic nature. Jain et al. [15] underlined the relevance of this study because of the relatively high metal quantities accumulated by the fungi in addition to the convenience of culturing. Some wood-rotting fungi are able to immobilize soluble metals into insoluble oxalate-crystalline forms [155], giving an interesting insight into a potential use of these strains in the metal bio-stabilization of contaminated sites.

To facilitate the synthesis, we recapitulated in Table 1 the studies that assessed the effects of inoculation of halophytes-associated microbes on plants submitted to metal and salinity stress.

Table 1. Referenced studies that measured the effects of inoculation of halophytes-associated microbes on plants subjected to metal(s) and salinity stress.

| Microbial Species Tested | Native Halophyte * | Metal(s) ** | Inoculated Plant and Its Metal Phytoremediation Potential in Parentheses | Type of Inoculum | Experimental Conditions | In Vitro Tested PGP and Metal Resistance Capacities | Effects of the Inoculation Compared to the Control | Metals' Remediation Capacity | References |
|---|---|---|---|---|---|---|---|---|---|
| *Aeromonas aquariorum, Pseudomonas composti, Bacillus* sp. | *Spartina densiflora* | Cu | *Spartina densiflora* (frequently recommended for phytoremediation) | Consortium of 3 bacteria, imbibition of seeds in 1 mL of bacterial culture | Plate assay. Evaluation of seed germination rate and fungal infection | N fixation, P solubilization, siderophores, auxins, high resistance to NaCl and Cu | Enhanced seed germination rate | Not evaluated | [127] |
| Two *Halobacillus* sp. strains and one *Halomonas* sp. strain | Mangrove rhizospheric soil | Co, Cd, Ni, Hg, Ag | *Sesuvium portulacastrum* (frequently recommended for phytoremediation) | Consortium of the 3 bacteria. Inoculation of cuttings with a 10 mL suspension of each of strain | Pot trial with cuttings cultivated in polluted sterilized sandy soil from coastal regions | IAA, P solubilization | Higher roots length and roots dry weight | Reduced metal contents in soil especially for Cd and Ni | [47] |
| As-resistant halophilic *Kocuria flava* and *Bacillus vietnamensis* | Rhizosphere of *Ceriops decandra* (mangrove species) | As | Rice (none) | Single-strain inoculation and co-inoculation of seedling rhizosphere with bacterial suspensions | Pot trial with seeds germinated in Soil Rite Mix | Absorption of As in bacteria, EPS, biofilms, siderophores, IAA | Higher root length, shoot length, dry weight, wet weight, and chlorophyll content | Not evaluated | [92] |
| *Pseudomonas composti, Aeromonas aquariorum, Bacillus* sp. | *Spartina densiflora* | Cu, Fe, K, Mg, Mn, Na, Zn | *Spartina densiflora* (frequently recommended for phytoremediation) | Consortium of 3 bacteria. Inoculation of seedlings with an amendment of 5 mL of each bacterial suspension | Pot trial with 2 soils from non-polluted and polluted marshes each supplemented with perlite (ratio 9:1) | Not evaluated | Higher root length and diameter, higher leaf water content | Higher metal accumulation in roots and leaves | [98] |
| *Bacillus methylotrophicus, Bacillus aryabhattai, B. aryabhattai, Bacillus licheniformis* | *Spartina maritima* | As Cu Pb Zn | *Medicago sativa* (frequently recommended for phytoremediation of non-saline soils) | Single-strain inoculation of germinated seeds with 5 mL of bacterial suspension | Plate assay in solid Fahraeus medium with different Cu concentrations | N fixation, P solubilization, siderophores, IAA, ACCD, biofilms | Higher root length, more lateral roots, increase of root hair formation | Not evaluated | [125] |
| *Vibrio neocaledonicus, Thalassospira australica, Pseudarthrobacter oxydans* | *Salicornia ramosissima* | As, Cd, Cu, Co, Ni, Zn, Pb | *Salicornia ramosissima* (recommended for phytoremediation, but scarce studies on this species) | Consortium of 3 bacteria, seedlings inoculated with 50 mL of bacterial suspension | Pot trial in soils from non-polluted and polluted marshes | N fixation, P solubilization, biofilms, ACCD, IAA, siderophores | Enhanced growth rate, higher number of branches, higher photosynthetic rate and functionality of PSII, and higher electron transport rate | Higher accumulation of As, Cd, Cu, Ni, Pb, and Zn, especially in roots | [30] |

**Table 1.** *Cont.*

| Microbial Species Tested | Native Halophyte * | Metal(s) ** | Inoculated Plant and Its Metal Phytoremediation Potential in Parentheses | Type of Inoculum | Experimental Conditions | In Vitro Tested PGP and Metal Resistance Capacities | Effects of the Inoculation Compared to the Control | Metals' Remediation Capacity | References |
|---|---|---|---|---|---|---|---|---|---|
| Several strains of *Halomonas* sp. | *Avicennia marina* | As | moderately salt-tolerant rice variety-*Jarava* (none) | Single and 6-strain consortiums. Germination test: seeds in bacterial suspensions. Pot trial: seedlings inoculated with 5 mL of each bacterial suspension | Effect on germination: plate assay. Pot trial: soil supplemented with NaCl, urea, muriate of potash, single superphosphate and NaAsO$_2$ | EPS, P solubilization, IAA, siderophores, NH$_3$ production, hydrogen cyanide, N fixation | Slightly enhanced germination; higher N$_2$ and PO$_4^{3-}$ contents in roots and shoots | Reduction of As(III) into a less toxic form [As(V)], reduction of As translocation | [136] |
| Consortia 1: *Kushneria marisflavi, Micrococcus aloeverae, Bacillus vietnamensis, Halomonas zincidurans;* Consortia 2: *Vibrio kanaloae, Pseudoalteromonas distinct, P. prydzensis, Staphylococcus warneri* | *Arthrocnemum macrostachyum* | As, Zn, Cu, Pb | *Arthrocnemum macrostachyum* (known as a promising candidate for phytostabilization of metals) | A 3-endophytic-strains consortium and a 3-rhizospheric-strains one. Seeds submerged with bacterial suspensions | Plate assay with a mixture of metals | Siderophores, N fixation, P solubilization | Acceleration of germination | Not evaluated | [45] |
| Two *Pantoea agglomerans* strains, *Bacillus aryabhattai* | *Spartina maritima* | As, Cu, Zn, Pb | *Spartina maritima* (frequently recommended for metal phytoremediation) | Single-strain inoculation and co-inoculation of sterilized seeds with 1 to 3 strains | Pot trial on seedlings, in collected sediments from non-polluted and polluted marshes | N fixation, IAA, P solubilization, siderophores | Higher germination rate, accelerated germination | Higher metal accumulation in roots only in the bacterial treatments with *B. aryabhattai* and with the 3 bacteria consortia | [46,156,157] |
| *Pseudomonas fluorescens* | Not known (from a manufacturer) | Cu, Zn, Cr | *Suaeda vera* (recently studied for its metal phytoextraction capacities) | One single strain. Dissolution of the inoculum in the irrigation water | Field trial on *S. vera* plantation. Cr−, Cu−, and Zn-contaminated soils | Not available | No effect on fresh weight | Increased metals' accumulation especially in roots | [158] |
| *Pantoea eucrina, Pseudomonas composti* | *Suaeda salsa* | Mn | *Suaeda salsa* (frequently recommended for metal phytoremediation) | Single strain directly added in the hydroponic solution | Hydroponic trial in Hoagland solution supplemented with 200 μM MnCl$_2$ | Not evaluated | Not evaluated | Enhanced Mn accumulation: Mn oxide precipitates at the root and leaves surface | [159] |

**Table 1.** *Cont.*

| Microbial Species Tested | Native Halophyte * | Metal(s) ** | Inoculated Plant and Its Metal Phytoremediation Potential in Parentheses | Type of Inoculum | Experimental Conditions | In Vitro Tested PGP and Metal Resistance Capacities | Effects of the Inoculation Compared to the Control | Metals' Remediation Capacity | References |
|---|---|---|---|---|---|---|---|---|---|
| Two *Bacillus pumilus* strains, *Azospirillum brasilense* | *Atriplex lentiformis* | Pb, Mn, Zn, Cu, As, Cd | *Atriplex lentiformis* (known for its metal-phytoextraction capacities) | Single-strain inoculation of seeds with microbead alginate inoculants added in the same planting hole | Pot trial with two substrate (acidic high-metal-content tailings and neutral low-metal-content natural tailings) supplemented with compost | N fixing and P solubilizing bacteria | Enhanced germination, root and shoot length, shoots, roots and total dry weight, root/shoot ratio, and number of leaves depending on the substrate | Not evaluated but the study aims to develop bio-assisted phytostabilization | [160] |
| *Glomus mosseae* | *Suaeda salsa* | Cd | *Suaeda salsa* (frequently recommended for metals phytoremediation) | Associated with biodegradable chelators nitrilotriacetic acid | Pot trial with soil from desert | Not evaluated (species largely used in the promotion of plant growth) | Reduced malondialdehyde concentrations in shoots; enhanced antioxidant defense, osmoregulation, and photosynthesis; promotion of specific bacterial communities | Enhanced Cd accumulation | [16,161] |

* From which strains have been isolated. ** Involved in the study cited in the "Reference" column.

## 4. Designing an Appropriate Inoculum for Phytoremediation or Phytostabilization

Until recently, the design of an inoculum was most of the time based on the selection of one microbial strain with various PGP traits, and only a few studies tried to elaborate a mixed inoculum (more representative of the initial microbiome) [162]. However, both methods (single strain or combination of several strains) can sometimes lead to "a depletion of metabolic diversity of the inoculum, whereas, in nature, cooperation within the microbial communities leads them to perform complex tasks" [162]. Kaminsky et al. [163] emphasized the difficulties of designing such an optimal inoculant because techniques are based on the artificial combination of strains that are most of the time selected based on an initial in vitro characterization. Even if these principles should enable us to precisely target potential microbial candidates, direct applications of the selected strains pose risks because their behavior can change in the field, as well as their effective PGP features. Moreover, screening microorganisms for their potential in microbial-assisted remediation leads to genetic trade-offs that could impair their natural metabolism or their capacity of roots colonization, for example [163]. Nevertheless, Yuan et al. [84] encouraged the combination of several microorganisms as inoculant, as it can increase soil microbial diversity and the phytoremediation yield through synergic mechanisms. Indeed, the study of Komaresofla et al. [95] showed an increase of beneficial effects of co-inoculation compared to single inoculation on the growth of *Salicornia* sp. inoculated with an endophytic *Staphylococcus* sp. strain and a rhizospheric one and subjected to diverse NaCl concentrations.

The success of phytoremediation greatly depends on rhizosphere microbiome functions and the way the plant interacts with its microbiome [164]. Furthermore, high amount and diversity of root exudates will favor competitive PGP microbes [164]. The soil microbiome can compete with the microbial inoculant for the same niche and alter the inoculant more or less rapidly [163]. When inoculants are successfully established, their spread into the environment must be carefully monitored because possible downstream impacts can occur, namely non-suitable changes in microbial diversity and/or biological invasion [163,165]. For example, the inoculation of *Alliaria petiolate* with two strains of *Glomus intraradices* (not native from the oldfield meadow soil used in the experiment), reduced the AMF local community diversity [166]. Similar effects have been found with a commonly used commercial inoculum strain of *Rhizophagus irregularis* in the study of Symanczik et al. [167]. Indeed, introducing this foreign AMF strain significantly reduced the abundance of native AMF species and the amount of extraradical mycelium. However, in a few cases, inoculation can increase the diversity of microbial communities [168]. Thus, the main challenge in the design of an inoculum is probably to find an optimal composition in order to have the least possible impact on the original plant microbiome while suitably contributing to host growth.

In addition, bioremediation strategies have to be designed according to the specificity of each polluted site, taking into account the metals' nature, soil characteristics, microbes, and plant communities [7], with many unpredictable parameters that evidently cannot be totally tested in the laboratory. This could be particularly the case in saline coastal areas where large fluctuations of salinity, humidity, and temperature can occur [122]. The availability of nutrients, the moisture content, the pH, the permeability, and the temperature of the soil matrix have to be characterized before the application of any bioremediation [169]. Moreover, vegetal cover and plant species diversity also modify soil parameters and shape microbial communities. In a field experiment of salt phytoremediation in a coastal area, Wang et al. [170] evaluated the microbial diversity in a bare soil and in four rhizocompartments (from the most distant part of the rhizosphere to the endosphere) of three plant species: *Gossypium hirsutum*, *Tamarix chinensis*, and *Lycium chinense*. At the end of the experiment, 12 years after, the three plant species had recruited significantly more bacterial and fungal species, whereas archaeal communities were the most diversified in the bare soil. Their results also indicated that the phytoremediated soil and bare soil differed regarding the electrical conductivity and soil moisture, thus affecting the composition of the microbial community. In addition, plant species differed in regard to the content of

total nitrogen, total carbon, and available potassium in the soil, which affected, in turn, the structure of microbial communities. According to Durand et al. [162], if we consider the plant microbiome as a superorganism, the microbiome is the part that would increase plants' adaptability to their environments.

Furthermore, we have to deal with a potential incompatibility between microbial candidates and associate the good partners, displaying complementary beneficial effects for plants [162]. The combination should increase the potential of the inoculum, and the selected strains should be cultivated separately in optimal growth conditions and be mixed at a second time in optimal proportions that enable both to be maintained in the rhizosphere [162]. Moreover, the selected microbes must be produced rapidly and in high quantity to be used at a large scale. Bashan et al. [137] distinguish four main types of inoculum, namely liquid, slurry, granular, or powder, that can be formulated in organic (for example, with peat, optionally associated with other organism(s) or biological matter(s), to improve the growth of some bacteria, such as dead mycelium, or lignite, charcoal, coir dust, or composts), inorganic (inorganic materials, natural polymers, or synthetic material), polymeric (alginate, agar, λ- and κ carrageenan, pectin, chitosan, and bean gum), and encapsulated (active microbes trapped in polymer matrix) formulations. Technical aspects include inoculation techniques that can be made by seed imbibition, watering the soil with a microbial suspension, applying inoculum powder on root system when preparing plants in greenhouse, or, more rarely, shoot spraying or through hydroponic plant culture [162]. Thus, selected strains must be compatible with such techniques.

## 5. Complementary Effects of Microbial Associations

The literature on PGP effects in saline environments revealed a disparity both in terms of in vitro/in vivo studies and between bacteria and fungi (compared to bacteria, fungi has received less attention). Oyetibo et al. [7] perceived a weaker interest in using fungi in bioremediation. According to the authors, the reason is that research on bioremediation "tends to disregard the ecological demands of fungi and often uses ecologically displaced organisms in competition with bacteria more suited to the polluted environment". Bacteria are easier to study, and numerous articles mentioned impressive capacities of bacteria to produce large quantities of EPS, to form biofilms, and to produce diverse phytohormones, as well as their capacities for high resistance to multiple stressors. However, fungi can also present many advantages. Fungal morphology is plastic and can adapt under metal stress. For example, they can produce pigments and thicken their cell wall by producing more chitin in response to metal stress [171]. Some AMF showed insensitivity to seasonality when colonizing roots under metal stress [141]. Some fungi, including AMF, can also produce organic acids that solubilize metals [172] and thus facilitate their phytoextraction in contrast with exopolysaccharides producing bacteria that rather stabilize metals. However, some bacteria have also been recognized for their metal-mobilizing properties [173,174]. Fungi easily can transport nutrients and contaminants for longer distances than bacteria [7]. Moreover, some bacteria can be inappropriate despite having interesting PGP capacities. For example, Bhaskar and Bhosle [128] investigated a *Marinobacter* strain which could produce high amount of EPS. However, the authors showed that these EPS represented a source of nutrients for *Hediste diversicolor*, a benthic polychaete, thus leading to a transfer of metals to higher trophic levels.

The synergic effects of the association between some fungi and bacteria sometimes help them to survive in stressful environments, thus preserving their respective positive effects on soil and plants. According to Sun et al. [21], facing constantly changing parameters in mixed polluted soils requires the combination of several methods to ensure an efficient remediation of metals. Saline soils are strongly influenced by winds, waves, tides, and temperature variations due to the opening of these environments and consequently fluctuate a lot in regard to their structural and chemical characteristics along time and space [122]. Their strong heterogeneity can compromise the establishment of microbes that need specific conditions to develop. Moreover, bacteria motility can be limited by

soil physical barriers (air-filled pores, dense aggregates, etc.) [7]. The use of fungi can be complementary to bacterial activities, as their highly branched mycelian filaments network can facilitate their spread in soils, as demonstrated by Wick et al. [175]. Reciprocally, Duponnois et al. [176] emphasized the beneficial effects of Mycorrhiza Helper Bacteria (MHB) on the establishment of ectomycorrhizal symbiosis. In an agricultural soil affected by salt and Cd, a greenhouse experiment was conducted on *Cajanus cajan* pretreated with a *Sinorhizobium fredii* strain and the AMF *Glomus mosseae* [177]. The phytotoxicity of Cd was increased, but the fungal symbiont was able to protect the *Sinorhizobium* nodules exposed to NaCl and Cd stress by mediating interactions between toxic ions and plants and reducing oxidative damages. They suggested an adaptation of the nodules of the *S. fredii* strain to mycorrhization, thus enhancing their salt and metal resistance [177].

Microbes can also be complementary between them. For instance, Teixeira et al. [178] showed an interesting way of designing a cadmium-resistant autochthonous bacterial consortia inoculum by selecting the cultivable fraction of the rhizospheric microbiome of *Juncus maritimus* and *Phragmites australis*. To do so, they cultivated them in their rhizosediments watered with estuarine water from the same place. They repeatedly added high Cd amounts and finally recovered a Cd-resistant consortium with sequential dilutions for each species and non-vegetated sediments. Their results revealed that the bacterial consortia enhanced various phytoremediation strategies among plant species and increased the phytostabilization capacities of *J. maritimus* and the phytoextraction capacities of *P. australis*. Mesa-Marín et al. [30] also showed positive effects of a bacterial consortium composed of three strains, namely *Vibrio neocaledonicus* SRT1, *Thalassospira australica* SRT8, and *Pseudarthrobacter oxydans* SRT15, on plant growth. The bacteria were isolated from the rhizosphere of *Salicornia ramosissima* and were selected for their PGP capacities, namely nitrogen fixation, phosphate solubilization, and biofilm-forming capacities, as well as their abilities to produce ACCD, IAA, and siderophores. They measured the bioaccumulation of Cu, Zn, As, Ni, and Pb and plant growth parameters in non-polluted and polluted estuarine soils with and without the bacterial inoculum. Inoculation increased the relative growth rate (32%), number of new branches (61%), net photosynthetic rate (21%), functionality of PSII, electron transport rate, and the intrinsic water-use efficiency (28%). The inoculated *S. ramosissima* plants accumulated more metals than non-inoculated ones, mostly in the roots (more than 1200 mg/kg when adding the five metals) due to plant biomass increment [30]. Berthelot and Leyval. [179] also discussed the beneficial effects of the combination of several fungi.

## 6. Bioremediation Applications Using Adapted Microbes

### 6.1. Microbially Assisted Phytoremediation: From the Lab to the Field Applications

Plant–microbe interactions are complex and greatly influence ecosystem functions. Numerous authors emphasize the importance of describing interaction mechanisms between plants and microbes to develop adapted, efficient, and sustainable bioremediation strategies [7,27]. Even though biological and bioengineering research generates more and more precious knowledge that ameliorates our understanding of the contribution of microbes to plants' stress-resistance capacity and adaptability, research is, most of the time, based on controlled or semi-controlled experiments on simplified conditions. A study from 2010 [50] asserted that, at this time, no in situ survey was conducted to concretely test the efficiency of the bioremediation of metals with halophilic microbes. In 2015, only 6.6% of the publications dealing with phytoremediation were based on field experiments [180]. Moreover, most of these publications are related to non-saline ecosystems. Indeed, extracting metals with plants or microbes seems easy to do in controlled conditions, and numerous articles make recommendations to foster bioremediation applications, but when confronted with the reality of the field, efficient bioremediation remains a difficult task. The influence of microbial communities on plant growth and their sensitivity to pathogens [27], as well as abiotic factors such as salinity, organic matter, pH, redox potential [1], and seasonality [181] determines the phytoremediation yield and success.

*6.2. Field Trials*

Even so, we gathered a few interesting field studies, including applications or possible applications to saline environments that bring an insight of the applicability of phytoremediation.

### 6.2.1. Potential of Applied Phytoremediation

Wan et al. [182] conducted a two-year phytoremediation project distributed along the Huanjiang River, in China, using the Cd hyperaccumulator *Sedum alfredii* and the Pb and As hyperaccumulator *Pteris vittata* to restore the functionality of these farmlands. They applied an intercropping technique in which hyperaccumulators were planted on middle contaminated zones, while cash crops were planted in the lowest contaminated zones. The results showed a significant decrease of available As, Cd, and Pb by 55.3%, 85.8%, and 30.4%, respectively, so that the produced food was safe again for the populations.

Ayyappan et al. [31] conducted an open-air experiment in large pot bags with the halophyte *Sesuvium portulacastrum*, a candidate for phytoremediation which has received much attention partly because of its metal-extraction capacities [71,72,183–185]. Seeds used for the experiment were harvested from native populations. Without introducing any microbial inoculant, encouraging results were obtained after 6 months, with high metal accumulation rates.

In their field experiment, Bareen and Tahira [80] obtained high capacities of metals' extraction in the leaves and roots of *Suaeda fructicosa* from a tannery effluent polluted site. Moreover, growth was augmented in the field compared with the pot experiment.

### 6.2.2. Bioaugmented Phytoremediation Trials

Guarino and Sciarrillo [186] showed that the inoculation of *Acacia saligna* and *Eucalyptus camaldulensis* with AMF and PGPRs to remove Cd, Pb, As, and Zn from an industrial polluted soil significantly contributed to enhanced plant growth and metal uptake. According to them, the benefits of the triple association plant–fungi–bacteria could limit metal spreads. On the contrary, in an 18-month field trial, Bissonnette et al. [187] revealed that AMF inoculant from *Glomus intraradices* did not increase metal extraction in *Salix viminalis* and *Populus generosa* from a non-saline soil contaminated with Cd, Zn, Cu, and Pb. Finally, a field study of Gómez-Garrido et al. [158] compared the impact of several metal-chelating organic acids and a *Pseudomonas fluorescens* bacteria on the metal phytoextraction capacity of the halophyte *Suaeda vera*. The results showed that *P. fluorescens* significantly increased the accumulation of Cu and Zn in the roots and stems in comparison with the control plants.

In view of these examples, such cost-effective and eco-friendly remediation techniques should be more and more concretely studied to assess field methods, taking into account all the parameters that influence the success of metal extraction or stabilization [28]. Halophytes-associated microbial strains that have already shown promising PGP effects should be further investigated in field trials to properly assess their applicability, a necessary step to be concretely used by professional operators.

## 7. Perspectives and Conclusions

The selection of microbial candidates based on their production of phytohormones, antioxidant enzymes, siderophores, or exopolysaccharides depends on the chosen phytoremediation strategy. Screening PGP microbes in in vitro conditions is the main method used to select relevant candidates but does not guarantee the proper establishment and persistence of inoculants in the field. Greenhouse and field trials highlighted the downsides of such approaches because indigenous microbial communities and fluctuant edaphic and environmental conditions interfere with root colonization of microbial inoculants and their efficiency [163]. As underlined by Yang et al. [188], these techniques can increase the complexity of the phytoremediation with many unpredictable effects and thus require multiple studies to prevent any long-term adverse effects in an ecosystem that is already weakened by metal contaminations. Moreover, phytoremediation, even in combination with microbes, is a slow process that is not adequate for urgent and rapid decontamination [24]. The com-

plexity of interactions between microbes–plant–soil and metals and the prior studies that must be achieved before a concrete application daunt remediation workers, as the process is often considered to be a series of time-consuming tasks with no guarantee of success [21]. Moreover, phytoremediation can sometimes be unsuitable when the contaminants are at a depth that the plant roots cannot reach, or when the pollution needs to be immediately remediated, as this technique requires repeated plantations [23,189]. However, in a lot of cases, microbes-assisted phytoremediation could be relevant, all the more so because most of the studies presented here have shown encouraging results, which should lead to an increase of the number of field studies in the near future and thus an improvement of the efficiency of biological techniques. The combination of several microbes has also been tested and could, in some cases, improve the application of these techniques. Moreover, bacteria and fungi can have synergic effects that can facilitate the establishment of each other while helping plants to stabilize or extract metals. Hence, field experiments with halophytes associated with microbes for the metal phytoremediation of saline soils should now increase to assess the applicability of large-scale projects.

The reason why such techniques take time to emerge could be related to the duration of field studies to assess their applicability and efficiency at the large scale of the inoculants, considering particularly the diversity of the environments and the stressful conditions. Moreover, even if the applicability is assessed, these biological solutions are still new and can also arouse suspicion for people according to the employed technique (metals stabilization that leaves contaminants in situ, suspicion of use of genetically modified microbes or plants, etc.) [190]. According to Wolfe and Bjornstad [190], "remediation decision making is a social process informed by scientific and technical information, rather than a science- or technology-driven process". The implementation of such a project requires population acceptance, which varies among cultures and individuals and depends on the perception of risks and values of the techniques [191]. Weir and Doty [191] measured the acceptability of people regarding phytoremediation of polycyclic aromatic hydrocarbons on 114 visitors of a park from a metropolitan area in the Pacific Northwest and found a high level of social acceptability of phytoremediation. Moreover, there is a lack of transfer of information and education to decontamination workers between the scientific and professional fields. A survey of Lachapelle and Montpetit [192,193] was conducted on 100 decontamination experts in Quebec, Canada, to evaluate the level of knowledge on biological decontamination technologies. It revealed a disappointing reality: more than 90% of the participants ranked their familiarity with phytoremediation under the score of 5/10, and more than half correctly answered less than two out of four basic questions. They noticed that educational background highly determines their sensitivity to such a question. In particular, engineers are the most representative of decontamination professionals and are the least educated in regard to phytoremediation concepts. The introduction of microbial inoculation and its complexity, especially in the context of saline areas, may add more difficulties to popularize these techniques. Within the EU Soil Strategy 2030, the European commission set objectives so that, "By 2050, soil pollution should be reduced to levels which are no longer expected to pose risks and which respect the boundaries our planet can cope with". They recommended employing biological remediation techniques for low-contaminated sites [194]. They highlighted the differences between European countries in terms of legislation, definitions, and methodologies for the risk assessment, remediation applications, and soil-contamination management and indicated the need to standardize these different skills across the European Union. A systematic analysis of publications [195] investigating the degree of research interests for each country on metal phytoremediation also showed that the numbers of studies and science projects on local metal-polluted sites are unequal between world countries. Moreover, Guarino and Sciarrillo [186] pointed out the fact that the overestimation of the pollution risk considering total metal content instead of its bioavailable fraction leads environmental politics in Italy to wrongly choose expansive conventional techniques when biotechnologies such as microbial-assisted bioremediation may suffice. Indeed, Summersgill [196] highlighted the differences in costs



between conventional techniques and bio-inspired alternatives. For example, in off-site methods, the incineration average cost is 885 euros/m$^3$, while average cost for off-site biological treatments is 167 euros/m$^3$. In on-site techniques, thermal treatments' average cost is 238 euros/m$^3$, while phytoremediation's min/max average cost is 122 euros/m$^3$. In situ bioremediation costs were even lower, with an min/max average of 73 euros/m$^3$ [196]. As another example, one of the largest phytoremediation projects was implemented in the region of Huanjiang, a highly polluted region contaminated by As, Pb, and Cd, and it cost only USD 37.7/m$^3$ [182]. This should draw the attention of policymakers to the choice of appropriate solutions for the treatments of metal-contaminated sites.

Hence, numerous studies reviewed here have demonstrated the interest of using one or several microbes to enhance the phytoremediation process in terms of reducing plant metal stress, improving plant growth, and extracting or stabilizing metal pollutants, particularly in saline areas. Researchers must continue to fuel knowledge on the species and site-specific applicability of these techniques and their predictability by conducting field trials when the efficiency of the plant–microbe system has already been assessed and by evaluating the long-term effects on the indigenous microbial and plant communities.

**Author Contributions:** Conceptualization, H.A., L.G., V.B.-S and P.B.; validation, H.A., L.G. and V.B.-S.; writing—original draft preparation, P.B.; writing—review and editing, H.A., L.G., V.B.-S. and P.B.; visualization, H.A., L.G., V.B.-S. and P.B.; supervision, H.A., L.G. and V.B.-S.; project administration, H.A., L.G. and V.B.-S.; funding acquisition, H.A., L.G. and V.B.-S. All authors have read and agreed to the published version of the manuscript.

**Funding:** This research was funded by the Foundation of the University of New Caledonia and the Laboratory of Excellence (LabEx) CORAIL (Coral reefs in the face of global change; EPHE-UNC A0 2022) and supported by the government of France (PhD. research grant from the French Ministry of Higher Education, Research and Innovation).

**Institutional Review Board Statement:** Not applicable.

**Informed Consent Statement:** Not applicable.

**Data Availability Statement:** Not applicable.

**Acknowledgments:** The authors would like to thank the Foundation of the University of New Caledonia and the LabEx CORAIL for their financial support.

**Conflicts of Interest:** The authors declare no conflict of interest.

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
