# Peer review of "Potential of Halophytes-Associated Microbes for the Phytoremediation of Metal-Polluted Saline Soils"

_applsci, doi:10.3390/app13074228_

Round 1

Reviewer 1 Report

Phytoremediation of metal pollutes soil is one of very important issues in environmental sciences. It is interesting and useful that the authors have reviewed using halophytes-microbes for phytoremediation of metal pollutes saline soils. In total, the MS was written sound. Hence, it is recommended to be published after some revising.

1. It is not clearly pointed out what kinds of species are more suitable for phytoextraction or phytostabilization in saline soils.

2. It is not presented the limitations of plants-associated microbes for the phytoremediation in saline soils.

3. it is not emphasized the future study directions for using halophytes-microbes for phytoremediation in saline soils.

Author Response

We thank you very much for the time you spent reviewing our review paper and for your very useful comments. Please find below the answers and explanations of the new added parts:

  1. It is not clearly pointed out what kinds of species are more suitable for phytoextraction or phytostabilization in saline soils

Precising this point should, in our opinion, need a more thorough discussion because from a study to another, some species can be indicated by authors for one to another removal strategy. Thus, we decided to give some examples and the references of the reviews that deal with this point instead of listing the species recommended in experimental studies.

See Line 167 : “[80]. In practice, the development of a phytomanagement strategy should require the definition of where metals will concentrate according to the field specificities and how to manage the remediated metals during or after the phytoremediation process. From this point of view, most of the publications evaluating the potential of halophyte spe-cies in metal phytoremediation distinguish the location(s) where metals are preren-tially accumulated in the plant, which generally determine the indication for phytoex-traction or phytostabilization. Manousaki et al. [34] mentioned as well another faculty of some salt excretors to proceed with phytoexcretion in supplement of phytoextrac-tion like Tamarix smyrnensis but this faculty should be further investigated. Naikoo et al. [36] presented as well a classification of the different halophyte species owing to the removal strategy that are frequently recommended for the phytoremediation of met-als, especially in India. For instance, species such as Atriplex halimus can be indicated for phytoextraction of Cd, Pb, Mg, and Zn while Arthrocnemum macrostachyum can be applied to Cd phytostabilization.”.

  1. It is not presented the limitations of plants-associated microbes for the phytoremediation in saline soils

We have now added some precisions on the limitations of phytoremediation and extensively of microbe-assisted phytoremediation.

See Line 705 : “As underlined by Yang et al. [188], these techniques can increase the complexity of the phytoremediation with many unpredictable effects and thus require multiple studies to prevent any long-term adverse effects in an ecosystem already weakened by metal contaminations. Moreover, phytoremediation even in combination with microbes is a slow process that is not adequate for urgent and rapid decontamination [24]. The complexity of interactions between microbes-plant-soil and metals and the prior studies that must be achieved before a concrete application daunt remediation workers as the process is often considered as a series of time-consuming tasks with no guarantee of success [21]. Moreover, phytoremediation can sometimes be unsuitable when the contaminants are at a depth that the plant roots cannot reach, or when the pollution needs to be immediately remediated as this technique requires repeated plantations [189,190]. However, in lot of cases, microbes-assisted phytoremediation could be relevant, all the more so  most of the studies presented here have shown encouraging results, which should lead to an increase of the number of field studies in the near future and thus an improvement of the efficiency of biological techniques.”

  1. It not emphasized the future study directions for using halophytes-microbes for phytoremediation in saline soils

We brought this information in the conclusion and summarized it as we already detailed the lack of information especially the field trials and we already argue why a follow-up of the bioaugmented-phytoremediation process was necessary to assess the long-term suitability and safety towards the ecosystem to be remedied. We have now completed these sentences.

See Line 776 : “Researchers must continue to fuel knowledge on the species and site-specific applica-bility of these techniques and their predictability by conducting field trials when the efficiency of the plant-microbes system has already been assessed and by evaluating the long-term effects on the indigeneous microbial and plant communities..”

Reviewer 2 Report

This is an interesting work and is well written. Minor revision is required before publication.

1.     Please emphasize the innovation of this work in the introduction.

2.     What is the difference and connection between this review and previous related reviews?

3.     More information on the development trend of future research should be added.

Author Response

We thank you very much for the time you spent reviewing our review paper and for your very useful comments. Please find below the answers and explanations of the new added parts:

  1. Please emphasize the innovation of this work in the introduction.

We now added a few lines to emphasize that this work is written in a practical view so that the context in which the use of this type of technique is relevant is clearer.

See Line 116 : “Given the magnitude of metal pollution on the world's coasts, this work provides reasoned perspectives on the application of microorganisms in phytomanagement techniques and highlights the obscure points of this type of applications that remain to be elucidated in scientific research. “

  1. What is the difference and connection between this review and previous related reviews?

We have now completed this point.

See Line 98 : “Furthermore, despite a substantial number of experiments conducted for this purpose, no specific review has, to our knowledge, focused on the use of halophyte-associated microbes as inoculants to improve the effectiveness of halophyte phytoremediation of metal contaminated saline soils. Indeed, research on PGP microbes was mainly reviewed in the context of metal contamination in non-saline sites [27,52–58]. Some reviews concerned applications to salt-affected sites using halotolerant bacteria for agriculture [41–44,59,60], but did not develop their potential for the improvement of metal phytoremediation. »

  1. More information on the development trend of future research should be added.

We brought this information in the conclusion and summarized it as we already detailed the lack of information especially the field trials and we already argue why a follow-up of the bioaugmented-phytoremediation process was necessary to assess the long-term suitability and safety towards the ecosystem to be remedied. We have now completed these sentences.

 See Line 777 : “Researchers must continue to fuel knowledge on the species and site-specific applicability of these techniques and their predictability by conducting field trials when the efficiency of the plant-microbe(s) system has already been assessed and by evaluating the long-term effects on the indigenous microbial and plant communities”